# Plant-Growth-Promoting Rhizobacteria Improve Germination and Bioactive Compounds in Cucumber Seedlings

**Laura-Andrea Pérez-García** [1,*], **Jorge Sáenz-Mata** [2], **Manuel Fortis-Hernández** [1], **Claudia Estefanía Navarro-Muñoz** [2], **Rubén Palacio-Rodríguez** [2] and **Pablo Preciado-Rangel** [1,*]

1   Tecnológico Nacional de México, Instituto Tecnológico de Torreón, Carretera Torreón-San Pedro km 7.5, Torreón 27170, Coahuila, Mexico
2   Facultad de Ciencias Biológicas, Universidad Juárez del Estado de Durango, Av. Universidad s/n, Col. Filadelfia, Gómez Palacio 35010, Durango, Mexico
*   Correspondence: dg21860128@torreon.tecnm.mx (L.-A.P.-G.); pablo.pr@torreon.tecnm.mx (P.P.-R.)

**Abstract:** Plant-growth-promoting rhizobacteria (PGPR) increase the germination percentage and the vigor of seeds, thus determining aspects for the efficient production of seedlings and the prompt establishment of crops in the field. In this study, the effect of the biopriming of rhizobacteria was evaluated: *Bacillus cereus* (KBEndo4P6), *Acinetobacter radioresistens* (KBEndo3P1), *Pseudomonas paralactis* (KBEndo6P7), and *Sinorhizobium meliloti* (KBEkto9P6) on some parameters such as the percentage of germination and vigor and the germination index, among others, as well as the synthesis of phytocompounds in the seeds of *Cucumis sativus* L. biopriming seeds significantly improved germination, the germination index, the vigor, the plumule and the radicle length, in addition to an increase in phytochemical compounds. The rhizobacteria KBEndo3P1 increased the germination percentage by 20%, the germination index by 50%, and the seed vigor by 60%, as well as the length of the radicle by 11%, and the plumule by 48% compared to the control, and the total phenols and antioxidants increased by 9% and 29%, respectively. Biopriming with plant-growth-promoting rhizobacteria increases germination, which allows for the possibility of more outstanding production of seedlings and a greater length of the radicle, thus increasing the efficiency in the processes of water and nutrient absorption and improving its establishment in the field. In addition, the production of phytocompounds enhances their response against any type of stress, making them a viable alternative in sustainable agriculture to increase cucumber yield.

**Keywords:** PGPR; seedlings; phytocompounds

## 1. Introduction

Cucumber (*Cucumis sativus* L.) is a plant of the Cucurbitaceae family widely cultivated worldwide in open fields and protected conditions [1]. These vegetables are consumed for their flavor and freshness, in addition to their nutritional contribution, since they are a source of fiber, carbohydrates, proteins, magnesium, iron, vitamins B and C, flavonoids, phenolic compounds, and antioxidants [2]. To ensure the productivity of this crop, a high-quality seed is required to provide adequate and prompt germination since this stage is one of the most sensitive to adverse conditions and directly influences the establishment, growth, and yield of the crop [3].

Currently, various alternatives have been studied to improve crop germination, to make them more environmentally friendly, and to reduce the use of biochemical compounds. There are studies about why PGPR are considered an option for sustainable agriculture due to the known importance of the mechanisms possessed by the different strains. However, strains continue to be isolated from stressful environments for their evaluation, especially in the germination stage since this stage is one of the most important stages to guarantee a high crop yield [4,5]. PGPR increase seed germination since they

contribute to the production of metabolites (siderophores and hydrocyanic acid), the synthesis of antibiotics, enzymes, and phytohormones (auxin, cytokinin, and gibberellic acid), and other associated activities such as a greater solubilization of phosphates in soil and root colonization [6–8], which increases the absorption capacity of nutrients and protection against phytopathogens, so PGPR represent a sustainable alternative for agricultural production [9]. Phytohormones in the germination process have an important role. In the case of gibberellins, in low concentrations, sterility can be observed in the seeds, and in high concentrations, it has a high potential as a stimulant in embryonic growth after dormancy is broken. In the case of auxins, such as indole acetic acid, they are growth regulators that can promote cell elongation and division, mainly produced in the apices of coleoptiles, with this hormone being essential for growth [10].

Presently, a wide variety of microorganisms have been characterized as PGPR, with them meeting different criteria. PGPR could be colonizers of plant roots in a significant way, promoting plant growth with the production of phytohormones and having tolerance to various physicochemical factors such as heat, drying, radiation, and oxidants, among other main criteria [11]. Strains of the genera include *Aeromonas, Agrobacterium, Allorhizobium, Arthrobacter, Azoarcus, Azorhizobium, Azospirillum, Azotobacter, Bradyrhizobium, Burkholderia, Caulobacter, Chromobacterium, Delftia, Enterobacter, Flavobacterium, Frankia, Gluconacetobacter, Klebsiella, Mesorhizobium, Micrococcus, Paenibacillus, Pantoea, Pseudomonas, Rhizobium, Serratia, Streptomyces,* and *Thiobacillus* [12,13], as well as the genera *Acinetobacter* that has the ability to synthesize phytohormones such as auxins and ethylene, in addition to indoleacetic acid. In the case of the *Bacillus* genus, it presents the production of cytokinins, auxins, and gibberellins, as well as the synthesis of secondary metabolites that confer antimicrobial activity [14]. The strains belonging to the genus *Pseudomona* have the characteristic of synthesizing indoleacetic acid, phosphate solubilization, nitrogen fixation, and producing phytohormones such as gibberellins and cytokinin, with them being key to plant development. Bacteria of the genus *Sinorhizobium* are characterized by being nitrogen fixers, by increasing the enzymatic production of 1-aminocyclopropane-1-carboxylic acid (ACC) or ACC deaminase from gene regulation when the plant is in unfavorable environments and hydrolytic enzymes that confer antimicrobial activity [15,16]. Considering that PGPR may have different mechanisms to improve yield in vegetables, the objective of this study was to evaluate the effect of new rhizobacteria in the germination stage and the effect on the production of phytocompounds in cucumber seedlings.

## 2. Materials and Methods

### 2.1. Vegetal Material

Cucumber seeds (*Cucumis sativus* L.) of the Poinsett 76 variety (Southern Star Seeds, S.A. de C.V.) were used as plant material for each treatment. First, the seeds were separated into batches of 10. Subsequently, the seeds were disinfected with a 10% (v/v) sodium hypochlorite solution for 5 min. Finally, the seeds were washed five times with sterile distilled water to remove the disinfectant solution and proceed to the biopriming [17].

### 2.2. Bacterial Strains

The PGPR were donated by the Microbial Ecology Laboratory of the Juarez University of the State of Durango; they come from the rhizosphere of different plants of the Chihuahuan desert that develop in conditions of high salinity or drought. The strains *Bacillus cereus* (KBEndo4P6) and *Sinorhizobium meliloti* (KBEkto9P6) were isolated from *Sartwellia* sp., KBEndo3P1 (*Acinetobacter radioresistens*) was isolated from *Peganum harmala*, and KBEndo6P7 (*Pseudomonas paralactis*) was isolated from *Tiquilia* sp. The activation of each of the microorganisms was previously carried out in conical tubes for 24 h at a temperature of 30 °C with shaking at 120 RPM (revolutions per minute). Subsequently, the inoculation of the bacterial culture was carried out in Erlenmeyer flasks with a volume of 250 mL Luria–Bertani culture medium (10 g Tryptone, 5 g NaCl, 5 g yeast extract, and 1000 mL of distilled water), at pH 7.3 with 200 µL of the strains a Allowed to incubate until

each strain had reached a cell concentration of $1 \times 10^9$ CFU mL$^{-1}$, which were used as the treatments [18].

### 2.3. Treatments

The strains that were selected to carry out the bioassays were *Bacillus cereus* (KBEndo4P6), *Acinetobacter radioresistens* (KBEndo3P1), *Pseudomonas paralactis* (KBEndo6P7), and *Sinorhizobium meliloti* (KBEkto9P6) As well as a control was used in which it consisted of not having any type of inoculation with rhizobacteria. In the experimental design, the distribution of the treatments consisted of a completely randomized design with five repetitions per treatment, considering a Petri dish as the experimental unit.

### 2.4. Germination and Growth Conditions

For the cucumber seeds, they were selected to be complete and of uniform size, five batches of 10 seeds each (n = 50) were placed, they were sterilized and washed 5 times with distilled water. First, biopriming was carried out using 50 mL of bacterial suspension of each strain for 60 min with shaking at 180 RPM in an orbital shaker (Lumistel M$^®$ ISO-45 Guanajuato, Mexico); the excess of each one of the samples was eliminated. Then, the treatments were allowed to dry in an oven at 35 °C for 48 h; the seeds were placed inside an artificial growth chamber (Yamato Scientific America$^®$, IC403 Santa Clara, CA, USA) at a temperature of 25 °C $\pm$ 2 °C with 60% of relative humidity and a photoperiod of 12:12 h light: darkness [19]. Once the imbibition period ended, 10 seeds were deposited on sterile filter paper moistened with 5 mL of sterile deionized water inside a Petri dish; subsequently, the seeds were placed carefully with the embryo located downwards. Subsequently, the Petri dishes were sealed and placed in an artificial growth incubator (Yamato Scientific America$^®$, IC 403) with a 12:12 cycle light period at 25 $\pm$ 2 °C with 60% relative humidity measured with (BOSCH Digital Multi-Scanner GMS120G, Mexico). Seed germination was quantified daily according to the International Seed Testing Association (ISTA, Switzerland 1999), analyzing the following growth parameters: germination percentage (G), germination index (GI), seed vigor (V), length plumule (PL), and radicle length (RL), as well as the fresh weight and dry weight of the shoots being captured up to the 7th day.

### 2.5. Parameters Evaluated during the Bioassay

The following parameters were evaluated for each treatment to make a relevant comparison. Germination percentage (G), counting the number of germinated seeds at the end of the established period (Equation (1)). This result is expressed as a percentage, the germination index (GI), which represents the number of germinated seeds due to the growth of the radicle, expressing this value in percentage of growth reached by the radicle during the bioassay (Equation (2)), where the percentage of the seed relative germination (SRG), the relative radicle growth (RRG), and the index of germination (IG) were included. For the percentage of seed vigor (V), carrying out the count after the 4th day of the germinated seeds, carrying out the first count, the second count on the 5th day, and so on, the count was carried out until the 7th day.

$$\text{G (\%)} = \frac{\text{Number of germinated seeds}}{\text{Total number of germinated seeds}} * 100 \tag{1}$$

$$\text{GI (\%)} = \frac{\text{SRG} * \text{RRG}}{100} \tag{2}$$

$$\text{V (\%)} = \frac{\text{Normal seed}}{\text{Total number of seeds}} * 100 \tag{3}$$

Plumule length (PL) was measured from the junction of the radicle with the hypocotyl to the base of the cotyledon, while the radicle length (RL) was measured from the base of the hypocotyl to the apex of the radicle. A digital vernier (Generic$^®$ DMC0144, Mexico) was used for the measurements, and the resulting values were expressed in centimeters.

Likewise, for the evaluation of the shoots, measurements of the fresh weight and dry weight were made, where for the first case, the samples were placed in a watch glass on an analytical balance, and the record was obtained. For the second case, the samples were placed in brown bags, with them being placed in a drying oven (Novatech S.A. de C.V.; Ohasus® 547A, Jalisco, Mexico) at a temperature of 72 °C for 24 h for subsequent data collection.

### 2.6. Indole Acetic Acid Production

The determination of the evaluation of the production of the following phytohormone, auxin, was carried out through the use of Salkowski's reagent, which results in the oxidation of indole compounds by ferric salts (Mayer, 1958). The IAA and its precursors are characterized by presenting a reddish color when in contact with said reagent. The intensity of the coloration of the reaction is directly proportional to the amount of IAA produced or one of its precursors [20], such as indole-3-pyruvic acid (IpyA) [21]. The appearance of the pink to fuchsia color is due to an oxidative reaction caused by the acid and a transamination that leads to the substitution of the amino group by Cl of $FeCl_3$, present in Salkowski's reagent (data not revealed). The bacterial strains were inoculated in 3 mL of LB broth added with one mg of L-tryptophan and incubated with constant shaking at 180 rpm for 15 days at a temperature of 28 °C. Subsequently, 3 mL of the bacterial suspension was recovered and centrifuged at 7000 rpm for 10 min. Two mL of the supernatant was extracted to which two drops of orthophosphoric acid were added and 4 mL of Salkowsky's solution (35% HCl with 0.5 M $FeCl_3$) was added. The samples were left to rest for 30 min at a temperature of 28 °C. Finally, the absorbance was read in a UV–Vis spectrophotometer (Thermo Scientific™, Genesys 20, Massachusetts, USA) at a wavelength of 530 nm, thus obtaining the data for its quantification. Previously, a calibration curve was made using different IAA concentrations (5, 25, 50, 75 and 100 μg/mL) with which the absorbance was compared to determine the IAA concentration in the samples.

### 2.7. Preparation of Extracts for Phytochemical Compounds

To obtain the phytocompounds, 2 g of fresh shoots were mixed in a volume of 10 mL of 80% ethanol with constant stirring at 120 RPM for 24 h. Subsequently, the extracts were centrifuged at 11,000 RPM for 10 min, where the supernatant of each sample was extracted for the following analyses [22].

### 2.8. Phytochemical Compounds

The content of total phenols was determined using a modification of the method of Folin– Ciocalteu [23]. For the samples used, 50 μL of each ethanolic extract obtained was taken; to the sample, 250 μL of Milli-Q water (MQ, Damstadt, DE) and 1500 μL of Folin–Ciocalteu reagent (1N) were added, shaken, and allowed to stand for 5 min. One Thousand Two Hundred μL of $Na_2CO_3$ (7.5% $w/v$) was added and shaken. The solution was allowed to stand for 2 h. The samples were quantified on a UV–Vis spectrophotometer (Thermo Scientific™, Genesys 20) at 760 nm. The standard was prepared with gallic acid, and the results were expressed in mg GAE 100 $g^{-1}$ fresh weight.

The total flavonoids were determined following the method of Zhishen [24]. For the sample, 250 μL of ethanolic extract, mixed with 1.25 mL of MQ water and 75 μL of $NaNO_2$ (5% $w/v$), was added, stirred, and allowed to react for 5 min. Next, 150 μL of $AlCl_3$ (10% $w/v$) was added, mixed, and allowed to react for 5 min. Subsequently, 500 μL of NaOH (1M) and 275 μL of MQ water were added and shaken vigorously. The samples were quantified in a UV–Vis spectrophotometer (Thermo Scientific™, Genesys 20) at 510 nm. The standard was prepared with quercetin dissolved in absolute ethanol, and the results were expressed in mg QE 100 $g^{-1}$ of fresh weight.

The total antioxidant capacity was measured by the in vitro 2,2-diphenyl-1-picryl-hydrazyl-hydrate (DPPH+) method [25]. For the sample used, a solution of DPPH+ (Sigma-Aldrich, St. Louis, MO, USA) mixed with ethanol was prepared at a concentration of

0.025 mg mL$^{-1}$; fifty μL of ethanolic extract was combined with 1950 μL of DPPH+ solution; after 30 min, the samples were quantified in a UV–Vis spectrophotometer (Thermo Scientific™, Genesys 20) at 517 nm. The results were expressed in μM equivalent in Trolox 100 g$^{-1}$ fresh weight.

### 2.9. Statistical Analysis

The results obtained were analyzed by analysis of variance (ANOVA) and comparison of means with Tukey's test ($p \leq 0.05$) using the statistical package SAS (Statistical Analysis System Institute) version 9.4. The normality of the data expressed as a percentage (germination percentage (G), germination index (GI), and seed vigor (V)) was verified with the Kolmogorov–Smirnov test and transformed by applying arcsine transformation and square root before the analysis of variance.

## 3. Results

### 3.1. Seed Germination

The evaluation of the different parameters they considered in germinating seeds using the different strains KBEndo4P6, KBEndo3P1, KBEndo6P7, and KBEkto9P6 positively increased germination in cucumber seeds (Figure 1) in which the increase in the size of the seeds can be observed. Plumule and radicle length of all treatments compared to the control. Just as it was observed that in the different treatments of the strains KBEndo3P1, KBEndo6P7, and KBEkto9P6, the increase in the number of secondary roots, in the case of the germination percentage, it could be observed that KBEndo3P1 presented the highest percentage compared to the control, as well as in the case of the variables evaluated for the germination index and vigor percentage, however, the treatments of the KBEndo4P6, KBEndo6P7, KBEkto9P6 strains were also observed to promote and improve the increase in the germination percentage, germination index, and vigor percentage (Table 1).

**Table 1.** Effect of rhizobacteria on the germination percentage (G), germination index (GI), vigor (V), plumule length (PL), radicle length (RL), fresh weight (FW), and dry weight (DW) in cucumber shoots.

| PGPR | G | GI | V | PL | RL | FW | DW |
|---|---|---|---|---|---|---|---|
| | % | | | cm | | mg | |
| Control | 76.6 e ± 0.51 * | 39.6 e ± 0.22 | 23.3 d ± 11.54 | 2.2 c ± 0.44 | 6.2 d ± 0.40 | 231.9 a ± 0.08 | 15.0 a ± 0.008 |
| KBEndo4P6 | 93.3 b ± 0.51 | 78.7 b ± 0.46 | 80.0 b ± 0.001 | 3.5 a ± 0.54 | 9.5 b ± 0.77 | 328.7 a ± 0.04 | 18.1 a ± 0.001 |
| KBEndo3P1 | 96.6 a ± 0.40 | 89.7 a ± 0.21 | 86.6 a ± 5.74 | 3.4 a ± 1.10 | 10.9 a ± 0.37 | 298.7 a ± 0.07 | 17.4 a ± 0.001 |
| KBEndo6P7 | 86.6 d ± 0.54 | 69.5 d ± 0.16 | 60.6 c ± 5.74 | 2.8 b ± 0.43 | 9.5 b ± 0.67 | 269.2 a ± 0.05 | 19.6 a ± 0.003 |
| KBekto9p6 | 90.0 c ± 0.54 | 72.3 c ± 0.26 | 86.6 to ± 5.74 | 2.8 b ± 0.001 | 8.7 c ± 0.78 | 311.4 a ± 0.05 | 18.1 a ± 0.006 |

* Data are shown as mean ± standard deviation (SD, n = 50). Values with the same letters in each column are similar according to Tukey's test ($p \leq 0.05$).

In particular, it can be observed that the KBEndo3P1 strain *Acinetobacter radioresistens* stimulates the synthesis of indole-3-acetic acid (IAA) production, and this compound, being an auxin, can in turn allow an increase in seed germination to occur, being that it favors cell division in the early stage of the embryogenic period [26,27]. There are several studies in which the importance of the production of indole acetic acid compound for seed germination is evaluated, obtaining favorable results that it increases by up to 15% in the percentage of germination compared to the control, for which it was observed that the production of this phytohormone favors seed germination [10]. PGPR promote plant growth from seed germination directly by modulating plant hormone levels in the sense that the KBEndo4P6, KBEkto9P6, and KBEkto9P6 strains obtained an increase in the germination percentage of 17%, 14% and 10%, respectively, with respect to the control. It can be inferred that the KBEndo3P1 strain synthesizes auxins and cytokinins, promoting growth and seed germination, as well as gibberellins continuing with the process of cell elongation of the embryonic tissues [11,28].

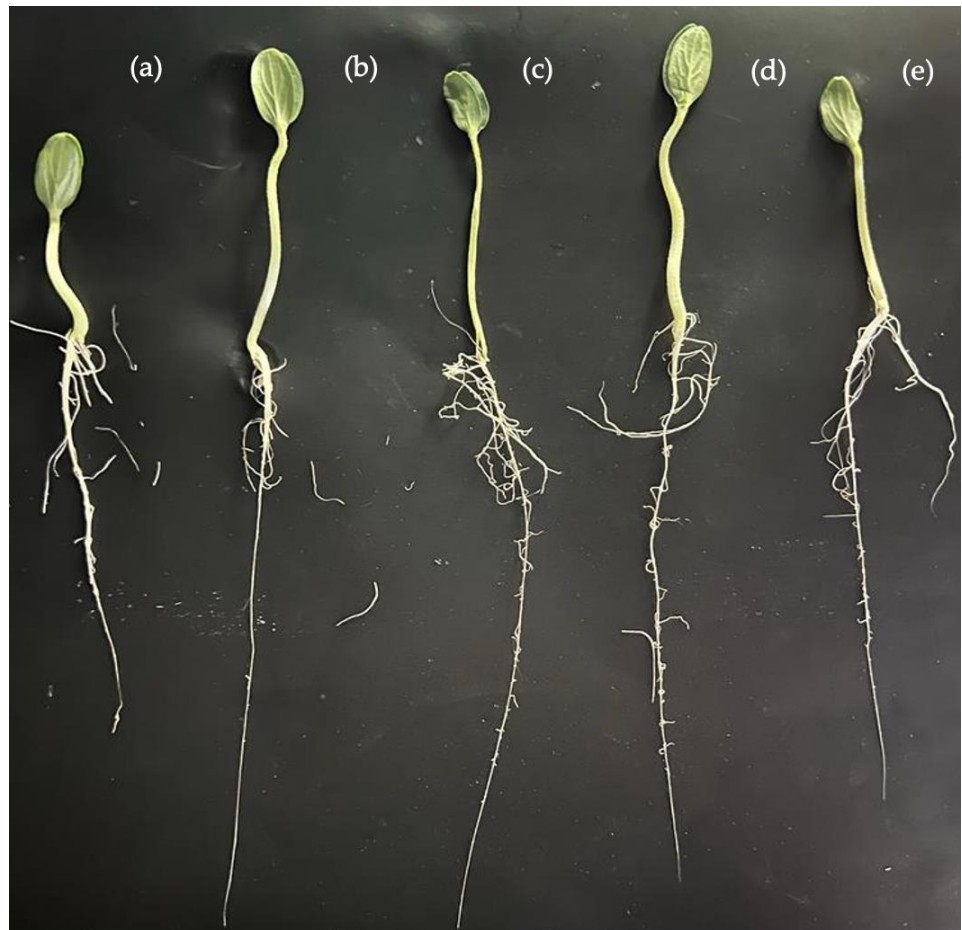

**Figure 1.** Cucumber seedlings at 7 days old, whose seeds were inoculated with plant-growth-promoting rhizobacteria: (**a**) control without any rhizobacteria inoculation, (**b**) *Bacillus cereus* (KBEndo4P6), (**c**) *Acinetobacter radioresistens* (KBEndo3P1), (**d**) *Pseudomonas paralactis* (KBEndo6P7), and (**e**) *Sinorhizobium meliloti* (KBEkto9P6).

*3.2. Production of Indole-3-Acetic Acid*

For the results of the evaluation of hormone synthesis, the production of a class of auxin that is indole-3-acetic acid (IAA) of the different strains was evaluated, in which it was obtained as a result that the strain with the highest production KBEndo3P1 strain is 0.392 mg mL$^{-1}$ of this phytohormone compared to the other strains, being the second strain KBEndo6P7 with a synthesis of 0.086 mg mL$^{-1}$, followed by KBEkto9P6 with 0.085 mg mL$^{-1}$ and finally, KBEndo4P6 with a synthesis of 0.076 mg mL$^{-1}$. Therefore, all the strains present the production of this compound that favors the germination process in the seeds (Table 2).

**Table 2.** Evaluation of the production of indole acetic acid IAA by the different rhizobacteria.

| PGPR | IAA mg mL$^{-1}$ |
|---|---|
| KBEndo4P6 | 0.076 ± 0.020 |
| KBEndo3P1 | 0.392 ± 0.048 |
| KBEndo6P7 | 0.086 ± 0.014 |
| KBEkto9P6 | 0.085 ± 0.011 |

*3.3. Phytochemical Compounds*

Biopriming with PGPR increased the biosynthesis of phytochemical compounds (Figure 2). The *Sinorhizobium meliloti* (KBEkto9P6) strain presents a 10% greater increase

in total phenols synthesis than the control. In the case of flavonoids, the treatments inoculated with the PGPR show that *Pseudomonas paralactis* (KBEndo6P7) and *Bacillus cereus* (KBendo4P6) present a higher synthesis of flavonoids, about 57% and 56%, respectively, compared to the other treatments. Regarding the antioxidant compounds, the highest percentage corresponds to the strain *Sinorhizobium meliloti* (KBEkto9P6) strain, with 30% compared to the control and a percentage greater than 18% with respect to the KBEndo6P7 strain, 20% higher against the strain KBEndo4P6 and 22% higher than the KBEndo3P1 strain, between the different treatments of the different strains used there is an increase in the production of phytochemical compounds with respect to the control improving the process of the germination stage.

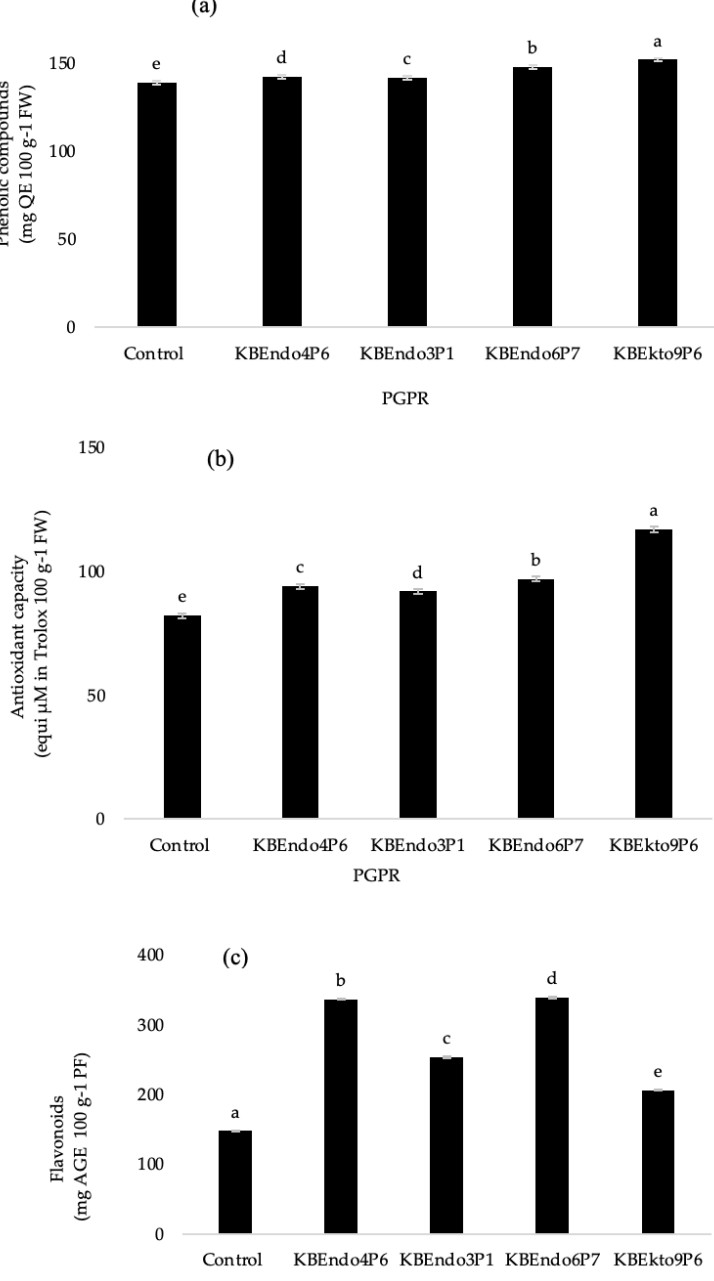

**Figure 2.** Effect of PGPR on the synthesis of phytochemical compounds. (**a**) Total phenolic compounds; (**b**) antioxidant capacity; and (**c**) flavonoids. Data are shown as the mean ± standard deviation (SD, n = 50). The average values in the columns with different letters differ statistically from each other (Tukey's test $p \leq 0.05$).

## 4. Discussion

The use of PGPR significantly increases the variables related to germination with respect to the control; the *Acinetobacter radioresistens* strain (KBEndo3P1) increased the germination percentage by 20%, the germination index by 50%, and the vigor in the seeds by 60%, as well as the radicle length by 48% and the plumule length by 11% with respect to the control. The use of PGPR induces greater germination, possibly due to the ability of PGPR to synthesize hormones such as indole acetic acid, gibberellic acid, and cytokinins, with them being substances that regulate cell division favoring seed germination. The treatments with the strains *Bacillus cereus* (KBENdo4P6), *Acinetobacter radioresistens* (KBENdo3P1), and *Pseudomonas paralactis* (KBENdo6P7) are those that present a more remarkable growth of root length which indicates that the *Pseudomonas* genus can increase the root length by up to 57% compared to the result obtained by the KBEndo6P7 strain, which increased the root length by 33% compared to the control [29]. In addition, there are reports that biopriming with PGPR positively improves the variables related to germination and growth uniformity under controlled growth conditions in a wide range of vegetables such as *Solanum lycopersicum*, *Allium cepa*, *Capsicum annuum*, and *Daucus carota* [30–32].

This result of the KBENdo3P1 can be attributed to the fact that this strain promotes the synthesis of phytohormones, such as indole-3-acetic acid promoting cell division [33]. In addition, PGPR such as *Acinetobacter radioresistens* (KBENdo3P1) and *Pseudomonas paralactis* (KBEndo6P7) enhanced the stimulation of different metabolisms for the synthesis of different plant hormones through the tryptophan biosynthesis pathway via the indole-3-pyruvate (IPA) pathway or the indole-3-acetamide (IAM) pathway for the production of IAA [34]. In addition, another mechanism through the oxidative pathway from a pyruvate molecule produces abscisic acid (ABA) that has the potential to control numerous aspects of the plant's life cycle, including seed dormancy, germination, and adaptive responses to the environment [35].

It is known that PGPR promote plant growth, improving the synthesis of different metabolites and phytochemical compounds, such as phenolic compounds, derivatives of benzoic acid, which can act as regulators in plant growth, with them being able to infer the increase in phenolic compounds in the germination stage [35–37]. Considering the above, the strains that were used KBEndo4P6, KBEndo3P1, KBEndo6P7 and KBEkto9P6 have the ability to stimulate the metabolic pathways responsible for oxidative stress, promoting the production of said compounds [38]. Studies have shown that the synthesis of these phytocompounds promote different physiological responses. In this case, it can be observed that the roots secrete flavonoids that act as signals in the rhizobacteria and these signals in turn influence the success of the plant-bacteria interaction, improving the colonization of the roots and regulating the expression of beneficial properties for the plant [31]. In addition, rhizobacteria promote the increase in the synthesis of antioxidants helping to counteract reactive oxygen species, preventing the oxidation of cellular components that leads to loss of functionality in plant cells [27,38]. Therefore, the use of these microorganisms improves the antioxidant synthesis capacity, it also be due to increased metabolic activity in the germination stage, and the synthesis of vitamins C and E, with variations in antioxidant activity, explained in relation to changes in the different phenolic compounds [38,39].

## 5. Conclusions

Biopriming with PGPR in cucumber seeds improves the parameters related to germination and the synthesis of phytochemical compounds in cucumber sprouts. The bacterial groups that promoted a more significant response in germination, vigor, and the development of plumules and roots, as well as the synthesis of non-enzymatic antioxidant compounds, were the strains: Acinetobacter radioresistens (KBENdo3P1), Pseudomonas paralactis (KBENdo6P7), and Bacillus cereus (KBENdo4P6). The use of PGPR is an alternative in sustainable agriculture by improving the germination and development of cucumber shoots, as well as greater production of phytochemical compounds, which allows early establishment in the field and subsequent growth of the plants.

**Author Contributions:** L.-A.P.-G.: conceptualization, methodology, data analysis, writing and original draft preparation, and editing; J.S.-M. and M.F.-H.: review, data analysis, and resources; C.E.N.-M. and R.P.-R.: conceptualization, methodology, and data analysis; P.P.-R.: conceptualization, review, data curation, research, and resources. All authors have read and agreed to the published version of the manuscript.

**Funding:** This research received no external funding.

**Data Availability Statement:** All data will be made available on request to the correspondent author's email with appropriate justification.

**Acknowledgments:** Laura Andrea Pérez García gives thanks for the financial support provided by the National Council of Science and Technology of Mexico (CONACYT) for doctoral studies and TECNM-ITT for the support and sponsorship of the project. The UJED provided the strains for their study, and the UAAAN provided the infrastructure to carry out the experimentation.

**Conflicts of Interest:** The authors declare that they have no competing interests.

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
