# Peer review of "Plant-Growth-Promoting Rhizobacteria Improve Germination and Bioactive Compounds in Cucumber Seedlings"

_agronomy, doi:10.3390/agronomy13020315_

Round 1

Reviewer 1 Report

Article title:

“Plant growth-promoting rhizobacteria improve germination and bioactive compounds in cucumber seedlings".

This article aimed to investigate the effect of the inoculation of the following rhizobacteria was assessed: Bacillus cereus (KBEndo4P6), Acinetobacter radioresistens (KBEndo3P1), Pseudomonas paralactis (KBEndo6P7), and Sinorhizobium meliloti (KBEkto9P6) on some parameters such as growth parameters and phytocompounds production of cucumber seeds. Regarding the manuscript, there are some points, that the authors should be to modify i.e.

Major comments:

1-      The manuscript idea is considered repeated, and the researchers must include what is new and highlight it in the objective.

2-      Change the term inoculation in the whole manuscript to the term dipping, soaking, or biopriming.

3-      The effect of the difference on the growth and productivity of plants was mentioned in the introduction section in lines 39-43, and it is inconsistent with the context of the experiment, so delete it.

4-      The aim of the manuscript at the end of the introduction section is not clear, it is general, so it must be rephrased again.

5-      In the Materials and Methods section, please mention the background from which these bacterial strains were selected in the subject of the study.

Minor comments:

1-   The writing style is poor and many grammatical errors should be corrected especially in (Abstract) which i could not follow the logic in the abstract such as "the length of the radicle and 11% of..". 2-      Line 66: Change Bacillus to Bacillus, Line 71: Change Sinorhizobium to Sinorhizobium, Line 128: Change 4th to 4th, Line 129: Change 5th to 5th, Line 170: Change FeCl3 to FeCl3, Line 302 and 303 change to Itallic. 3-   Rewrite again the text from line 102 to 109" Un clear". 4-   Re-correct the equations in line 132 and 136 into English language. 5-   Replace part No. 2.5 with part No. 2.6 in Materials and Methods section 6-   Line 226 – 235: There are the discussion in Results section, move it to discussion section.  

Author Response

Dear reviewer, thank you for your comments on the manuscript "Plant growth-promoting rhizobacteria improve germination and bioactive compounds in cucumber seedlings" have been very helpful to improve our paper, below I list each of the observations addressed, hoping that they will be evaluated and accepted by you.

  • The manuscript idea is considered repeated, and the researchers must include what is new and highlight it in the objective.
  • Currently, various alternatives have been studied to improve crop germination, be more environmentally friendly, and reduce the use of biochemical compounds. There are studies about why PGPR is considered an option for sustainable agriculture due to the known importance of the mechanisms possessed by the different strains. However, strains continue to be isolated from stressful environments for their evaluation, especially in the germination stage since this stage is one of the most important to guarantee a high crop yield. 
  • Change the term inoculation in the whole manuscript to the term dipping, soaking, or biopriming.

  • The word inoculation was changed to the word biopriming throughout the manuscript
  • The effect of the difference on the growth and productivity of plants was mentioned in the introduction section in lines 39-43, and it is inconsistent with the context of the experiment, so delete it.
  • The lines was removed from the manuscript
  • The aim of the manuscript at the end of the introduction section is not clear, it is general, so it must be rephrased again.
  • Considering that PGPR may have different mechanisms to improve yield in vegetables, the objective of this study was to evaluate the effect of new rhizobacteria in the germination stage and the effect on the production of phytocompounds in cucumber seedlings.
  • In the Materials and Methods section, please mention the background from which these bacterial strains were selected in the subject of the study.
  • The PGPR were donated by the Microbial Ecology Laboratory of the Juarez University of the State of Durango; they come from the rhizosphere of different plants of the Chihuahuan desert that develop in conditions of high salinity or drought. The strains Bacillus cereus (KBEndo4P6) and Sinorhizobium meliloti (KBEkto9P6) were acquired from Sartwellia mexicana, KBEndo3P1 (Acinetobacter radioresistens) from Peganum harmala and KBEndo6P7 (Pseudomonas paralactis) from Tiquilia sp. The microorganisms were previously activated in conical tubes for 24 hours at a temperature of 30 ºC. 250 mL flasks were inoculated in Luria-Bertani culture medium (10 g Tryptone, 5 g NaCl, 5 g yeast extract, and 1000 mL of distilled water), at pH 7.3) with 200 μL of the strains and left incubate, until reaching a cell concentration of 1×109 CFU mL-1, which were used as treatments [18].
  • The writing style is poor and many grammatical errors should be corrected especially in (Abstract) which i could not follow the logic in the abstract such as "the length of the radicle and 11% of..". 
  • The rhizobacteria KBEndo3P1 increased the germination percentage by 20%, the germination index by 50% and the seed vigor by 60%, as well as the length of the radicle by 11%, and the plumule by 48% to control and total phenols and antioxidants increased by 9% and by 29%, respectively.
  • Line 66: Change Bacillus to Bacillus, Line 71: Change Sinorhizobium to Sinorhizobium, Line 128: Change 4th to 4th, Line 129: Change 5th to 5th, Line 170: Change FeCl3 to FeCl3, Line 302 and 303 change to Itallic
  • In all these observations the changes were made in the manuscript. 
  • Rewrite again the text from line 102 to 109" Un clear". 
  • Strains used for the bioassays were Bacillus cereus (KBEndo4P6), Acinetobacter radioresistens (KBEndo3P1), Pseudomonas paralactis KBEndo6P7), and Sinorhizobium meliloti (KBEkto9P6) and used a control without bacteria. In the experimental design, the distribution of the treatments consisted of a completely randomized design with five repetitions per treatment, considering a Petri dish as the experimental unit.

  •  Re-correct the equations in line 132 and 136 into English language
  • Each equation was changed to English in the manuscript
  • Replace part No. 2.5 with part No. 2.6 in Materials and Methods section
  • The sections were changed in the materials and methods section
  • There are the discussion in Results section, move it to discussion section.  
  • We made the paragraph change from the results section to the discussion section

Reviewer 2 Report

1. Microbial strains used in the study must be appropriately mentioned in the manuscript.

2. The study can include introducing stress condition like moisture regime or salinity and evaluate the potential impact of introduced PGPR towards seedling germination

2. The antimicrobial property studies of the bioactive compounds can further strengthen the role of PGPR towards improved germination in seedlings.

3. Abbreviations for Indole acetic acid (IAA) to be presented correctly in the manuscript.

4. Inclusions of relevant and current citations to be included in the manuscript.

5. Overall improvements required in English language of the draft.

Author Response

Dear reviewer, thank you for your comments on the manuscript "Plant growth-promoting rhizobacteria improve germination and bioactive compounds in cucumber seedlings" have been very helpful to improve our paper, below I list each of the observations addressed, hoping that they will be evaluated and accepted by you.

Point 1:  Microbial strains used in the study must be appropriately mentioned in the manuscript.

Response 1: The strains were corrected in the manuscript by changing them to italics

Point 2: The study can include introducing stress condition like moisture regime or salinity and evaluate the potential impact of introduced PGPR towards seedling germination

Response 2: In this article, we chose not to include information about how PGPR can improve the effect of germination under stress conditions because our research group is working on the experimental part of these kinds of stress.

Point 3: The antimicrobial property studies of the bioactive compounds can further strengthen the role of PGPR towards improved germination in seedlings.

Response 3: In this article, we chose not to include information on which kind of PGPR contains antimicrobial effects and can these effects improve germination because we don't have jet any results about this interaction but our research group is working on the experimental design to evaluate which of these strains can have antimicrobial compounds like bacteriocins in the case of the Bacillus genus

Point 4: Abbreviations for Indole acetic acid (IAA) to be presented correctly in the manuscript.

Response 4: For the results of synthesis, the production of indole-3-acetic acid (IAA) of the different rhizobacteria was evaluated, obtaining as a result that the strain with the highest production of this compound is the strain KBEndo3P1compared to the other strains. However, all have a production of this compound favoring the germination process in seeds (Table 2).

Table 2. Evaluation of the production of indole acetic acid IAA by the different rhizobacteria

PGPR

IAA

mg ml -1

KBEndo4P6

0.076 ± 0.020

KBEndo3P1

0.392 ± 0.048

KBEndo6P7

0.086 ± 0.014

kbekto9p6

0.085 ± 0.011

Point 5: Inclusions of relevant and current citations to be included in the manuscript

Response 5: Monalisa, S. P., & Roy, S. (2022). Chapter-5 Effect of Biopriming with Plant Growth Promoting Rhizobacteria (PGPR). Chief Editor Dr. RK Naresh, 73.

 Miljaković, D., Marinković, J., Tamindžić, G., Đorđević, V., Tintor, B., Milošević, D., ... & Nikolić, Z. (2022). Bio-priming of soybean with Bradyrhizobium japonicum and Bacillus megaterium: Strategy to improve seed germination and the initial seedling growth. Plants11(15), 1927.

Point 6: Overall improvements required in English language of the draft.

Response 6: Throughout the manuscript, grammatical changes and revisions of the entire language in English were made.

Round 2

Reviewer 1 Report

Accept in present form